# AIM5LA: A Latency-Aware Deep Reinforcement Learning-Based Autonomous Intersection Management System for 5G Communication Networks

**DOI:** 10.3390/s22062217

**Published:** 2022-03-13

**Authors:** Guillen-Perez Antonio, Cano Maria-Dolores

**Affiliations:** Information Technologies and Communication Department, Universidad Politécnica de Cartagena, 30203 Murcia, Spain; mdolores.cano@upct.es

**Keywords:** 5G latency forecasting, 5G vehicular communication, autonomous intersection management, autonomous vehicles, connected autonomous vehicles, cooperative autonomous driving, intelligent transport systems, intersection traffic management, latency-aware driving, unsignalized intersection

## Abstract

The future of Autonomous Vehicles (AVs) will experience a breakthrough when collective intelligence is employed through decentralized cooperative systems. A system capable of controlling all AVs crossing urban intersections, considering the state of all vehicles and users, will be able to improve vehicular flow and end accidents. This type of system is known as Autonomous Intersection Management (AIM). AIM has been discussed in different articles, but most of them have not considered the communication latency between the AV and the Intersection Manager (IM). Due to the lack of works studying the impact that the communication network can have on the decentralized control of AVs by AIMs, this paper presents a novel latency-aware deep reinforcement learning-based AIM for the 5G communication network, called AIM5LA. AIM5LA is the first AIM that considers the inherent latency of the 5G communication network to adapt the control of AVs using Multi-Agent Deep Reinforcement Learning (MADRL), thus obtaining a robust and resilient multi-agent control policy. Beyond considering the latency history experienced, AIM5LA predicts future latency behavior to provide enhanced security and improve traffic flow. The results demonstrate huge safety improvements compared to other AIMs, eliminating collisions (on average from 27 to 0). Further, AIM5LA provides comparable results in other metrics, such as travel time and intersection waiting time, while guaranteeing to be collision-free, unlike the other AIMs. Finally, compared to other traffic light-based control systems, AIM5LA can reduce waiting time by more than 99% and time loss by more than 95%.

## 1. Introduction

Autonomous Vehicles (AVs) are becoming an integral part of our society due in part to recent advances in Artificial Intelligence (AI) and technology improvements, which enable safer, easier, and more reliable travel, along with a large reduction in the number of injuries and fatalities caused by human error. AVs are designed to drive themselves on highways and freeways with great ease, but urban roads are a challenge yet to be solved. Therefore, the next step is to develop AVs that are able to drive in challenging traffic conditions, such as busy streets with pedestrians, cyclists, emergency vehicles, traffic lights, multi-signals, intersections, etc., and make them capable of adapting to different traffic conditions and user behaviors, thus providing greater safety for all traffic actors.

An approach to addressing the challenge of urban traffic is using Autonomous Intersection Management (AIM) [1]. An AIM is a centralized control system, located at an intersection and usually referred to as an Intersection Manager (IM), that coordinates the behavior, states, and actions of all AVs (speed, acceleration, steering, path, etc.) crossing the intersection to eliminate accidents due to human error, thus improving traffic flow. These AIMs assess a control policy based on the status of all vehicles and road users; to obtain their status, AIMs communicate with the vehicles to obtain multiple data such as geographical position, current speed, acceleration, orientation, route, etc. In addition, AIMs can integrate data from external sensors [2,3] such as cameras, noise sensors, pollution sensors, etc. Once all the data is obtained, AIM evaluates the control policy and obtains the action (next target speed, acceleration, route, etc.) that each AV should take to ensure safety and maximum flow. Figure 1 shows a representation of an AIM. Note that AIM refers to the control algorithms of the AVs and IM to the node in charge of the communication between the AVs and the AIM.

In this context, the control of AVs using AIM requires knowledge of the state of all AVs to be controlled to make the right decisions, as well as AVs cooperating to achieve the same goals. A cooperative control system must be aware of the state of all AVs at the intersection and adapt its strategy in a time-limited manner. In addition, it must have the necessary foresight to be able to anticipate potential problems and cope with uncertainties. To the authors’ knowledge, most research studies on cooperative control in AIM focus on the design of control strategies, and do not consider (or mostly assume ideal conditions) the communication latencies inherent in wireless communication systems between AVs and the IM.

Due to the lack of works studying the impact that the communication network can have on vehicle control and safety, this paper presents a novel latency-aware deep reinforcement learning-based AIM used for AV control in the 5G communication network, called AIM5LA. It was decided to use the 5G communication network given the advantages it offers for vehicle control using Ultra Reliable Low Latency Communications (URLLC). This is the first latency-aware AIM trained fully by Multi-Agent Deep Reinforcement Learning (MADRL). When we say it is latency-aware, we mean that AIM5LA adapts the control of the AVs to the inherent latency of the 5G network, thus providing safety and smooth traffic flow. If communication latency were not considered, a small delay in control could cause a vehicle to receive an order later than desired, leading to undesirable situations, including collisions.

Although AIM5LA is derived from an AIM system implemented in previous work [4], AIM5LA has several improvements. These enhancements are:(i)Taking into account the previous latencies of all AVs using a deep neural network based on a recurrent neural network to encode the latencies of the other AVs;(ii)Predicting the latency that each AV is going to experience in the next timestep, using a Transformer-based deep neural network, to adapt the control of the AVs;(iii)Consider the number of AVs to be controlled simultaneously as an internal parameter to adapt the control according to the number of AVs;(iv)Propose the set of messages as well as the communication protocol for AIM5LA implementation.

The rest of the article remains as follows: In Section 2, we present the related work on AIM and focus especially on latency-aware AIMs. In Section 3, we present the proposed AIM5LA mechanism as well as the necessary message protocol. The simulations performed as well as the testbed used are described in Section 4. In Section 5, the results obtained are analyzed and discussed. Finally, Section 6 concludes and proposes future work.

## 2. Related Works

AIMs represent a very suitable approach to solving traffic problems because of the great versatility they offer. They have been extensively studied and have demonstrated an ability to reduce accidents and increase traffic flow by decreasing congestion and delays at urban intersections.

The first AIMs were proposed by Dresner et al. in [5,6,7,8,9]. These AIMs were the first Query-Based (QB) methods applied to AVs control at intersections. The operation of the proposals presented by Dresner et al. was based on the vehicle attempting to cross the intersection requesting the IM if it could pass (or not), indicating its estimated time of arrival. The IM could accept the request if there were no collisions with other vehicles based on the Estimated Time of Arrival (ETA). Otherwise, the IM would send a Denial of Request (DoR), and the VA would have to slow down and ask again later. This approach was based on the First-Come-First-Served (FCFS) policy and outperformed traditional traffic light control systems in low traffic flows (<500 veh/h).

Since FCFS presented several serious problems that could lead to risky situations [10], Huang et al. proposed in [11] a QB alternative that allowed grouping vehicles according to their status and allowed the IM to indicate the recommended deceleration speed when there was a DoR. The results showed a reduction in time loss of about 85%, as well as a reduction in fuel consumption of 50%.

Other FCFS improvements were those developed by [9,12], through which it was possible to create platoons of vehicles traveling in the same direction, increasing the intersection flow. The results obtained improved the flow of vehicles by ×2 and reduced the time loss by 85% compared to traditional traffic light control techniques.

An alternative to FCFS was suggested by [13], in which the right-of-way policy was calculated based on the fastest vehicle arriving at the intersection, called Fast First Service (FFS). The findings were superior to traffic light-based techniques, allowing a reduction in hard-braking.

Other control techniques, in addition to FCFS, use mathematical optimizations to determine the passing priority policy [14,15,16,17,18]. Although the results obtained with these techniques were similar to those obtained previously in work using FCFS, the high complexity of large intersections makes a real-time control solution unfeasible.

A control policy different from FCFS was proposed by Carlino et al. in [19], where the proposed policy was auction-based. The vehicles that bid the most to cross were the ones that could pass through the intersection first, creating a market economy. Although the results shown were superior to those of the compared algorithms, the problems derived from this economy could generate vehicle starvation, which could suffer indefinitely long waiting times, as well as generate inflation, discrimination, etc.

Some works use Reinforcement Learning (RL) to address this approach. For example, the works proposed in [10,20] used RL to obtain the vehicle passing priority policy, demonstrating excellent performance when compared to other control algorithms and reducing waiting time by more than 80%.

Another work using Deep Reinforcement Learning (DRL) was presented in [4], where DRL was used to obtain an advanced vehicle control policy. In this case, the IM used DRL to find the optimal actions to be taken by the vehicles (the speed to which the vehicles should adapt at each time interval) when approaching (and within) the intersection. In that proposal, it was necessary to use training acceleration techniques such as curriculum learning [21] and Twin Delayed Deep Deterministic policy gradient (TD3, [22]) to reduce the training time because the training scenario could be considered highly complex. The results obtained showed excellent performance in multiple metrics, such as a reduction in time loss of up to 95%, reductions in pollutant gas emissions of around 50%, or reductions in fuel and energy use of 25%.

As we have seen in this short review of the state of the art, most of the work focused on AIM is based on communication between VAs and IMs but does not consider the latency required for communication between them. There is a small set of works that have taken it into account [23,24,25,26]. Of note is that of Zheng et al. [24], where they presented a delay-tolerant intersection management protocol that considered the network delay for vehicular control. Their findings revealed that the presented method outperformed traditional traffic signal systems when traffic was low (<360 veh/h) or asymmetric (the flow of the north-south branches was three times greater than that of the east-west branches). However, when traffic was medium-high, or when the traffic was symmetrical, the result shown by their proposal did not surpass that obtained with traffic lights.

Another interesting work that accounts for Round-Trip communication Delay (RTD) was presented by Andert et al. in [25]. In this case, RTD was estimated by message exchange and followed a similar approach to FCFS. The results showed that they could improve the flow of a single intersection compared to FCFS by 26%. An improvement of the previous work was presented by Khayatian et al. in [26]. In this case, they improved the synchronization process between IM and VAs by message exchange and enhanced the decision-making algorithm. They used FCFS as the method for passing-priority decision-making. The results obtained showed a reduction in the positioning error by more than 15%, increasing the throughput by 36% compared to other AIM techniques, as well as a complete reduction in crashes. However, both AIMs were based on fixed mathematical control rules that do not allow for learning an advanced and adaptive control policy. Moreover, the proposed system was only tested in scenarios where the traffic density was extremely low (<300 veh/h).

It is clear that the works on AIM so far have focused on improving vehicular flow as well as reducing lost time, but very few have considered the influence of latency inherent in wireless communication systems. As a result, we believe that there is still a dearth of work addressing the study of latency in AIMs, particularly using advanced approaches such as multi-agent DRL, in which the system can learn autonomously without the need for any control rules and obtain advanced control techniques. By doing so, it could offer higher security and the possibility to learn from the environment under control to perfectly adapt its behavior to the conditions and properties.

## 3. Our Proposal

AIM5LA is based on *adv*.RAIM [4]. *adv*.RAIM implements an advanced cooperative AV control system for urban intersections using AI algorithms. Specifically, it uses DRL’s TD3 algorithm along with other algorithms designed to accelerate training, such as PER. *adv*.RAIM offers an advanced control policy in an *ego* way, i.e., it processes all AVs recursively, calculating the speed at which each AV should travel during the next control interval as a function of the *ego*-vehicle’s state and the other “collaborating” vehicles’ states. This process is repeated periodically every timestep for all AVs (250 ms).

*adv*.RAIM is included within the collaborative problems and follows the mechanism of centralized training and centralized execution, which allows for the obtaining of the better performance of the experiences performed by the different actors (vehicles) in the environment (intersection). The experiences are centralized in the IM, being the IM in charge of executing the control policy for all the controlled AVs. The experiences that each vehicle sends to the IM are composed of the status of each vehicle (geographic position, speed, route, lane, etc.). When the IM collects all the experiences of all the AVs to be controlled, it has a complete and detailed view of the whole intersection. On these experiences, the DRL algorithm is applied, in which, based on the different actions that the vehicles can perform (speed during the next time interval), it calculates the optimal speeds to be followed by the AVs during the next time interval to minimize the travel time without causing collisions between the AVs.

In addition, to solve the problem of independence of the number of AVs in the control algorithm, *adv*.RAIM incorporates a recurrent neural network, a Long Short-Term Memory (LSTM). This LSTM encodes the representation of the characteristics (state) of the other AVs with which the *ego*-vehicle collaborates into a fixed-size output variable (256) and captures the long-term spatial and temporal dynamics of the traffic conditions in the network. The output of this LSTM was concatenated with the parameters (state) of the *ego*-vehicle. The main drawback is that *adv*.RAIM does not take into account the latency inherent to the communication systems. Communication is necessary for the data exchange (status) between AV and the AIM system in charge of calculating *adv*.RAIM. Although originally *adv*.RAIM enables accident-free control, this was because it was tested in a traffic simulator without communication latency.

Due to the lack of works exploring this problem using DRL, in this work, we proposed to evolve *adv*.RAIM to take into account the latency of the 5G communication network and thus improve control in real future scenarios where mobile wireless communication protocols such as 5G will be used. The new proposal, called AIM5LA, includes the following improvements and modifications compared to *adv*.RAIM:-Incorporates the previous latency experienced by the *ego*-vehicle to be controlled in the previous interval.-Includes a novel latency prediction module that predicts the latency experienced by the *ego*-vehicle during the next control interval, based on a Transformer deep neural network and the history of latencies experienced, as well as the number of AVs to be controlled simultaneously.-Considers the latency experienced by other AVs at the intersection using an encoder LSTM network.-Finally, we propose the set of messages, the time intervals between each message, as well as the communication protocol, to implement AIM5LA.

In our humble experience, this is the first latency-aware AIM trained entirely by DRL. When we say it is latency-aware of the communications network, we mean that AIM5LA is able to learn a control policy that can deal with changes in communications latencies and thus internally model the behavior of AVs based on their latency, and therefore adapt the control of these AVs to avoid collisions.

Without considering communication latency, a delay in control could cause an AV to receive a command later than desired, leading to undesirable situations, including collisions.

Once the above modifications have been included, a representation of the AIM5LA architecture can be seen in Figure 2. This figure shows the observation space of AIM5LA.

The latency predictor module is composed of a Transformer-based network [27] that predicts the latency that the *ego*-vehicle will experience in the next time interval based on the history (*h* in Figure 2) of latencies experienced by that vehicle and the number of AVs in the intersection (as later discussed, the number of AVs with a 5G communication module affects latency). We believe that the enormous advances provided by Transformers-based networks in countless areas may outperform alternative neural network forecasting networks such as LSTM or GRU [28,29,30].

To use AIM5LA, we propose a protocol message exchange for communication between the AIM module and the AVs, as well as to calculate the latency in the communication channel. This protocol includes the following messages:

-*Probe*: The *probe* message is sent by the IM of each intersection and contains IM-specific information (communication characteristics, geographical position, etc.), as well as a timestamp used to estimate the latency in the communication channel. This timestamp is the internal time of the IM just before sending the message. This message is sent periodically with a sending period such as to allow safe control of the AVs, but without saturating the communication channel. A suitable period is 250 ms, based on the control intervals of other works [20,31] and our experience with *adv*.RAIM [4] development. This means that every 250 ms (four times per second), vehicles update the action to be performed (speed, acceleration, route, etc.). As we have seen in previous work, a shorter update period (e.g., 100 ms) increased the processing load, limiting the real-time processing of the system. On the other hand, a longer update period (e.g., 500 ms) made the system less safe, as too much time elapsed between two updates, leading to control crashes.

The other message timing is adjusted to meet with the 250 ms update period. Each vehicle, after receiving the *probe* message from the IMs, checks whether its geographical position and route match with the IM that should control it, responding to this message with a control *request* message.

This periodicity enables sequential and temporally separated control of all AVs without saturating the processing or communication channel, as well as safe and fast control with sufficient anticipation.

-*Request*: These messages are sent by AVs to the IM to indicate that they want to be controlled during the next control interval (250 ms). To avoid channel saturation and delays due to overlapping with other AVs that want to send the control *request* message, AVs wait a random time *t* to send the control *request* message. The random time *t* is used to reduce the probability of overlap of sending messages when multiple AVs attempt to access the 5G URLLC channel resources and thus remove this overlap from the latency calculation. After performing a preliminary set of experiments in the Simu5G simulator [32], it has been found that the optimal values of the waiting time *t* follow a continuous uniform distribution with a minimum value of 0 ms and a maximum value of 50 ms. This interval minimizes the probability of overlap with other AVs, considering the use case of an intersection; furthermore, it allows a compromise so that all control *requests* can be processed in real-time. The control *request* message includes a vehicle identifier and the timestamp resulting from adding the random time *t* for sending the control *request* message to the timestamp captured from the IM *probe* message. In addition, it includes the internal parameters necessary for the cooperative control of the AVs, which include geographical position (x, y), speed, acceleration, branch of the intersection (north, east, south, west), lane in which it travels (left, center, or right), route it intends to follow (through, left turn, or right turn), type of vehicle, physical characteristics (width, length, weight), and technical characteristics (maximum speed, maximum acceleration, etc.). With this message, the AIM can estimate the latency existing in the wireless communication channel. When the IM receives the message from the AV, it only has to compare the timestamp included in the message with the internal timestamp, so it calculates the latency existing in the complete IM-AV–IM path. In addition, a history of the computed latencies allows for the assessment of an estimate of the jitter, which is also an interesting parameter. Finally, thanks to the random time *t*, the time that each vehicle takes to gather internal parameters (position, speed, trajectory, etc.) is eliminated in the calculation of the communication channel latency.

-*Action*: This message is sent from the IM to each controlled AV and indicates the *action* to be performed at that moment until the next control interval. This message may include speed, acceleration, steering wheel position, geographical route to follow, speed and acceleration profile, etc. The calculation of this *action* already takes into account the possible latency of the wireless communication channel as well as the influence of the other AVs at the intersection. In this case, AIM5LA predicts the speed to be followed by the *ego*-vehicle during the next control interval.

The message exchange diagram can be seen in Figure 3. In that figure, we can observe that, from the AIM’s point of view, after sending the *probe* message (at time T1), the AVs are given enough time to respond, taking into account the communication latency, as well as the random time *t* (t_V1, t_V2, and t_V3) that each vehicle waits to respond. The time allotted to listen to control *requests* (control *request* messages) is 100 ms. This value takes into account the up to 50 ms that an AV may wait to respond to the *probe* message and the latency that may exist in the system in the extreme case of a saturated scenario [32,33]. In order not to leave any request out, the maximum limit was set at 100 ms.

Consequently, the AIM can calculate the average latency $Latency_{T_{i+1}}^{Vx}$ for vehicle Vx during the following time interval $T_{i+1}$, as shown in Equation (1), where $T\_Request^{Vx}$ is the time instant at which the control *request* message is received from vehicle Vx, $t^{Vx}$ the random time t that vehicle Vx has waited to send the control *request* message, and $T_{i}$ the time instant at which the AIM sent the *probe* message.
(1)$$Latency_{T_{i+1}}^{Vx}=\frac{T\_Request^{Vx}-t^{Vx}-T_{i}}{2}$$

From that moment on, the AIM starts processing the requests to obtain the best actions that AVs should perform in the next control interval. As a result, it will have the most up-to-date information from all vehicles, as well as from other data sources such as cameras, people counters, sensors, etc. In addition, the AIM can already estimate the latency of the communication between each of the AVs using the timestamps included in the *request* messages and the internal captured timestamps of these messages. This latency is also fed into the vehicle control decision-making process implemented in this work using AIM5LA, in addition to forecasting the latency behavior. This stage was assigned a maximum time of 75 ms. This time is based on the time required to perform the inference of the deep learning models employed, as well as the remaining time of the self-imposed control period (250–100 ms of the *probe* + *request* messages). The actual inference time is expected to be much less than this time due to the use of concurrent computing techniques, pruning, knowledge distillation, and half-precision floating-point operations.

Finally, the *actions* to be performed/maintained during the next interval are sent consecutively to each AV. For this process, we assign a maximum of 75 ms, which is the remaining time of the control interval period (250 ms of *control interval period*–100 ms of *probe* + *request*–75 ms of *processing*). Then, the process is started again, but this time with the time instant T2 (=T1 + 250 ms [*probe message control interval*]).

## 4. Testbed and Experiments

In this section, the testbed is developed for AIM5LA training as well as the simulated scenarios.

### 4.1. Testbed

To evaluate the performance of AIM5LA and the proposed message protocol, several scenarios were designed. The first scenario was used to train AIM5LA to find an advanced control policy that takes into account the wireless communication latency. The second scenario aimed to show the performance of AIM5LA in a scenario never seen before by the algorithm during training to demonstrate that AIM5LA can find robust control despite confronting unknown traffic and latency situations. In addition, before carrying out the experiments in these scenarios, the latency predictor module had to be optimized in a scenario using the Simu5G 5G computer simulation tool [32].

AIM5LA derives its control policy from DRL [34] and makes use of other advanced techniques and algorithms such as TD3 [22], PER [35], and curriculum learning [21] to accelerate training and maximize learned knowledge.

To simulate the wireless communication channel and protocol, the 5G simulator Simu5G 1.2.0 [32] was used, along with OMNeT++ 5.6.2 [36] and INET 4.2.2 [37]. Simu5G is an OMNeT++ library for the end-to-end performance evaluation of 5G networks. It was decided to investigate latency in the 5G mobile communication protocol since it is a widely developed and researched protocol with ultra-low expected latency (1 ms). It focuses on assessing Vehicle-to-Network (V2N) use cases over the 5G New Radio (NR) and follows the Rel-16 specification to model the 5G NR radio access network and the data plane. In this case, the IM was simulated as a 5G gNodeB node and the vehicles, as 5G devices (NrUe–NrCar), were configured to offer low latency and throughput similar to that offered to the URLLC use case.

For the simulation of vehicles, their behavior, and urban roads, SUMO 1.8.0 [38] was used together with TraCI [39] for vehicle control using AIM5LA.

For the development of AIM5LA, as well as all the deep learning algorithms, Python 3.8.10 was used together with the deep learning framework PyTorch 1.9.0 with CUDA [40]. As for the hardware, an Intel 8-core/16-thread processor (i7-11700k) and an Nvidia RTX 3080 graphics card were used. Table 1 summarizes the parameters used in the different simulators.

### 4.2. Experiments

First, the latency prediction module was optimized. We called it Experiment #1. We did the optimization by simulating a simple 2-lane, traffic light-controlled 60-s cycle intersection with an increasing number of vehicles. This is because it has been shown in previous works [32,33] that the latency experienced by 5G network devices depends on the number of active devices. Therefore, simulations were performed in the previously described scenario, with 1, 4, 16, 64, 128, and 256 vehicles. Simulations were run 10 times, and the vehicles followed random routes in each simulation, and the communication latency between the vehicle (nrCar in Simu5g) and a 5G base station (gNodeB in Simu5g) was measured. The gNodeB was located at the edge of the intersection, at a height of 20 m. This intersection was controlled by a traditional traffic light system. For the latency calculation, the message exchange described above was implemented in Omnet++, where the base station gNodeB sent the *probe* message periodically (every 250 ms), and when this message was received by the vehicles, the vehicles responded with a control *request* message after a random time *t* that followed a uniform distribution with a minimum of 0 ms and a maximum of 50 ms, as shown in the previous section.

For training the latency predictor module, the gNodeB (acting as IM) was configured to send *probe* messages every 250 ms. The scenario had a duration of 300 s, and the first 250 s of the simulation were used as the training dataset and the last 50 s as a testing dataset, obtaining an 83%/17% split of the dataset. The forecaster predicts the latency experienced based on previous latencies, i.e., it considers a previous temporal window (*h* in Figure 4) to forecast the expected future latency behavior. This temporal window was optimized during training to find the optimal window size that obtained the best performance in latency prediction. In addition, since the latency experienced could depend on the number of AVs at the intersection, it was decided to include this feature in the latency forecast.

Finally, the latency forecast module included a neural network based on Transformers, with an output of 1 hidden neuron. The output of this network was concatenated with the number of simulated AVs variable and these variables were used as input to a deep neural network composed of 3 Fully Connected (FC) layers, with 64, 16, and 1 neuron in each layer, respectively, with ReLU as activation functions in the hidden layers. The neuron in the last layer is responsible for providing the latency forecast output during the next control interval. A representation of the architecture of the prediction module can be seen in Figure 4.

Once the latency forecast module was optimized, the AIM5LA control algorithm could be optimized. The results of this optimization can be seen in the following section.

To optimize the AIM5LA control algorithm, another scenario was developed, namely Experiment #2, consisting of an intersection with an AIM in charge of controlling the vehicles crossing that intersection. The intersection had four 200-m branches (north, east, south, and west), 3 lanes per direction, left turns, straight ahead, and right turns were allowed. AVs were controlled when within 100 m of the center of the intersection. A representation of this scenario can be seen in Figure 1.

To obtain a wide variety of situations that challenge the control algorithm, the simulated flow pattern was set as shown in Figure 5a. The vehicle flow pattern shown indicates the number of simulated vehicles per lane per simulated hour on the North (N), South (S), West (W), and East (E) branches. The flow is symmetrical on the opposite branches (NS and WE).

The metrics, optimized by the DRL algorithm, were the number of collisions and time loss of vehicles due to intersection congestion, with the reward function followed by the DRL algorithm being as follows:-−100 when there was an accident;-+100 when a vehicle crossed the intersection without an accident;-*−control interval* (−0.25) at each simulation interval to encourage optimization of time loss and crossing as quickly as possible.

These reward signals are determined in the original paper [4]. The metrics analyzed in the training scenario were the waiting time of the vehicles, the number of accidents, and the global reward obtained by the DRL algorithm in each simulation. The optimization process was performed three times, and the results showed the moving average of the mean and the standard deviation.

A final scenario was composed of a Manhattan network of 100 intersections (10 × 10) to show the performance of the AIM5LA algorithm, namely Experiment #3. At each intersection, there was an AIM in charge of controlling the vehicles crossing it. The intersections were separated by 250 m, with a similar configuration to the training scenario (4 branches, 3 lanes, left, straight, and right movements allowed). In addition, to illustrate singular situations, a different flow behavior was simulated, shown in Figure 5b. This scenario shows the capabilities of AIM5LA against situations never seen before during training. A representation of this scenario can be seen in Figure 6.

The metrics analyzed in the test scenario were diverse because SUMO provides a complete set of metrics after each simulation. These metrics were time loss due to congestion at intersections, number of collisions, average waiting time, various pollutant gas metrics (CO_2_, PMx), and fuel and electric consumption. Although all the metrics showed are not directly optimized, they are indirectly optimized through the overall optimization process.

The vehicle configuration used is shown in Table 2. As can be seen, there were several types of vehicles, with different physical dimensions and characteristics that made them unique, which are available in [41].

To compare the performance of AIM5LA with other algorithms, we also included the performance evaluation of *adv*.RAIM [4], the AIM algorithm proposed by Andert et al. [25], and two preliminary versions of AIM5LA, using only the latency of the *ego*-vehicle at the previous time instant (called AIM5LA_v0.1) and AIM5LA_v0.1 + the latency forecaster module (called AIM5LA_v0.2). The use of the preliminary versions of AIM5LA (AIM5LA_v0.1 and AIM5LA_v0.2) serves as an ablation study, allowing us to show the importance of each module, as well as to isolate their contribution. The results obtained with traffic-light based control algorithms have been incorporated as well; particularly, a fixed time control (FX) system with different cycle lengths (30, 60, and 90) and an intelligent advanced traffic light control technique (iREDVD) [2] that proactively adapts the traffic lights to reduce congestion. The fixed traffic light control system assigns a fixed green time for each branch of the intersection where the passing priority (green light) is given. This passing priority is cyclically passed through all branches with a fixed cycle time (in our case, 30, 60, and 90 s were analyzed).

## 5. Results

In this section, we show and discuss the results.

### 5.1. Experiment #1—Forecast Module Optimization

After performing the simulations, the average latency, as well as its standard deviation as a function of the number of simulated vehicles, can be seen in Figure 7. These results show that the latency value strongly depends on the number of vehicles and can increase by up to ×10 if we increase from 1 to 256 vehicles simulated simultaneously, which corroborates the expectations. This increase in latency can be caused by a variety of factors, including communication channel saturation, a decrease in available bandwidth, overlapping communication requests with other devices, and so on. In addition, a summary of the results can be found in Table 3.

The temporal behavior of the latency for a random vehicle in each of the simulated scenarios can be seen in Figure 8a. In this figure, the dependence of the number of AVs on the latency experienced can be seen. Furthermore, it is clear that the greater the number of devices simulated simultaneously, the greater the latency variance (~jitter). This is important when predicting the latency behavior; therefore, this variable of the number of simultaneous AVs is important to consider for the latency prediction module because it influences both the average latency value and its variation.

Once we had simulated the behavior of the AVs in OMNeT++ using the Simu5G library and obtained the experienced latency, we had a dataset that could be used to train the latency prediction module. For the division of the dataset, the first 250 s of latencies were used as the training dataset, and the remaining 50 s were left as the test and validation dataset. This division of the dataset can be seen graphically in Figure 8b below. This figure shows the latency behavior for one vehicle in the 4-vehicle scenario, showing the division of the dataset into training (<Message #1000, blue line) and test (≥Message #1000, red line).

The results of the training and testing can be seen in Figure 9. In it, the Root Mean Square Error (RMSE) [42] metric as a function of the historical latency window size (*h* in Figure 4) has been represented. In addition, the results are shown both when the number of vehicles parameter was considered (w/nvehs) and when it was not considered (w/o nvehs) as an input parameter to the deep neural network of the latency forecast module.

From these results, the optimal size of the historical latency window (*h*) was about 10 samples, since it makes the least error with the smallest window size. Finally, when we considered 10 previous samples for latency prediction at the next time instant, as well as the number of simultaneous vehicles, the mean value and standard deviation of the RMSE results were 0.4551 ± 0.0264 in the training set and 0.3702 ± 0.0301 in the test dataset.

This means that the latency predictor module has a high accuracy in forecasting the latency of any vehicle during the next control interval just by considering a history window of 10 latency samples. With this predicted latency, the other AIM5LA modules will have to adapt the control, taking into account the time it will take for the vehicle to react due to latency.

For a visual representation of what these results represent, in Figure 10, we illustrate the behavior of the predictor module during the training and the testing phases. Indeed, these results were remarkably accurate, being able to strongly predict the latency value in both the training and test datasets. Figure 10 shows the result for the 4-vehicle scenario (Figure 10a) and 128-vehicle scenario (Figure 10b).

### 5.2. Experiment #2—Training AIM5LA

Once the latency forecast module was trained, AIM5LA was trained. The training results can be seen in Figure 11. This figure shows the following metrics: training evolution of the reward (Figure 11a), time loss (Figure 11b), and number of collisions (Figure 11c). The mean value (solid line) and standard deviation (shaded area) of the three runs are depicted. The curves shown have been smoothed using a moving average of size 100 samples for visual clarity.

The training results show the great performance offered by AIM5LA, being able to completely reduce collisions from about simulation 10^6^ onwards. After this, AIM5LA continued to optimize the control policy, adjusting the control of the AVs to reduce time loss. This is reflected in the reward metric, showing a large improvement when the number of collisions is reduced and, when achieved, the reward improvement is reduced but always increases. The optimization of AIM5LA required 21 days (approx. 500 h) of simulation and training for the three runs, using an Intel 8-core/16-thread processor (i7-11700k) and an Nvidia RTX 3080 GPU.

### 5.3. Experiment #3—Benchmarking AIM5LA

Finally, after the training process, AIM5LA was compared with other AIM and traffic-light based protocols in the test scenario described above. The results are included in Table 4.

AIM5LA presents an excellent performance, being the only AIM protocol able to reduce accidents in the test scenario. Looking at other metrics such as time loss, waiting time, pollution, or fuel consumption metrics, AIM5LA can achieve a performance similar to other AIM techniques such as the one proposed by Andert et al. [25] or *adv*.RAIM [4]. However, it should be noted that the performance offered by AIM5LA is limited by the elimination of collisions, therefore obtaining a similar result gives a higher value to our proposal.

On the other hand, comparing AIM5LA with the v0.1 version of AIM5LA (which only considers the previous latency of the *ego*-vehicle), we see that there are some collisions that AIM5LA_v0.1 is not able to eliminate due to the instability of the channel in the latency behavior, as well as due to the influence of other AVs. Last, when compared to AIM5LA_v0.2 (that is, AIM5LA_v0.1 plus the latency prediction module), the result is significantly improved. However, there is still a small number of collisions that are completely eliminated when using the final version of AIM5LA.

In general, AIM5LA achieves great performance in all the analyzed metrics, showing a reduction in time loss of up to 92.25% when compared to other traditional traffic light control (FX) algorithms, or a reduction of more than 99.52% in waiting time. When comparing AIM5LA with other advanced traffic light control techniques (iREDVD [2]), the results show a significant reduction in metrics such as time loss (84.51%), waiting time (99.25%), fuel consumption (35.67%), or pollutant gas emissions (48.50%).

Finally, looking at the performance of AIM5LA compared to other AIM techniques such as *adv*.RAIM and the proposal of Andert et al., we conclude that the performance offered is similar to these techniques, but where our proposal stands out is in the metric of the number of collisions.

It is worth noting that, in the work where *adv*.RAIM was presented, there were no crashes due to communication latency because this was not considered in the simulation setup. Nevertheless, when testing its performance in a real scenario such as this one, where there is communication latency in the 5G mobile communication protocol, it is observed that there are indeed crashes due to this lack of consideration of latency (see Table 4), indicating the importance of this factor.

## 6. Conclusions

Cities are a challenging environment for Autonomous Vehicles (AVs). In particular, they are complex and chaotic environments that require advanced control to ensure maximum safety. One of the solutions proposed by the scientific community to control AVs at intersections is Autonomous Intersection Management (AIM). AIM is based on communication between AVs and a control entity located at each intersection to obtain a joint control policy that improves the intersection flow. However, the vast majority of AIMs ignore the latency that may exist in the communication channel, which poses a great safety vulnerability and a high risk of accidents.

The main contribution of this work is the development of a latency-aware AIM for 5G communication networks integrally trained by Multi-Agent Deep Reinforcement Learning (MADRL). Our proposal uses a MADRL algorithm with centralized training and centralized execution and a network latency forecasting module to obtain an advanced control policy to ensure control security and reduce traffic congestion. In addition, our proposal uses a novel latency predictor module that employs a Transformer-based deep neural network. This is the first latency-aware AIM fully trained by MADRL. When we say latency-aware, we mean that our proposal adapts the control of the AVs to the inherent latency of the 5G network, thus providing traffic security and fluidity. The system was named AIM5LA, and beyond considering the latency history experienced, AIM5LA predicts future latency behavior to provide better security and improve traffic flow through the latency predictor module. AIM5LA was compared with other AIMs [25] as well as other advanced traffic light-based control techniques such as iREDVD [2]. Unlike prior AIM proposals, the results demonstrated AIM5LA’s strong robustness against collisions, fully eliminating them.

First, the latency prediction module was able to obtain surprising results, achieving an RMSE of less than 0.4. This allowed it to predict with high accuracy the latency that an AV would suffer during the next control interval. When comparing AIM5LA with traditional traffic-light based control techniques, AIM5LA is able to reduce the time loss by 92%, as well as the waiting time by more than 99%. Furthermore, compared to advanced adaptive traffic light control techniques such as iREDVD, AIM5LA was able to reduce time loss by 84.51% and achieve a significant reduction in other metrics such as time loss (84.51%), waiting time (99.25%), fuel consumption (35.67%), or pollutant gas emissions (48.50%). Moreover, observing the results obtained with AIM5LA during the ablation study (v0.1 and v0.2), it is possible to observe the contributions of the different modules developed, as well as the importance of considering them in different metrics such as travel time, number of collisions, etc. More specifically, only the full version of AIM5LA can eliminate crashes in the test scenario. The most important lessons learned during this work included the importance of the latency prediction module to eliminate collisions, as well as the importance of using centralized training and end-to-end multi-agent deep reinforcement learning algorithms, with excellent action space exploration, to obtain high efficiency in future AIM systems.

In future work, the forthcoming 6G architecture will be considered, as well as the inclusion of unmanned aerial nodes to reduce communication latency and analyze performance in a real 5G system.

## Figures and Tables

**Figure 1 sensors-22-02217-f001:**
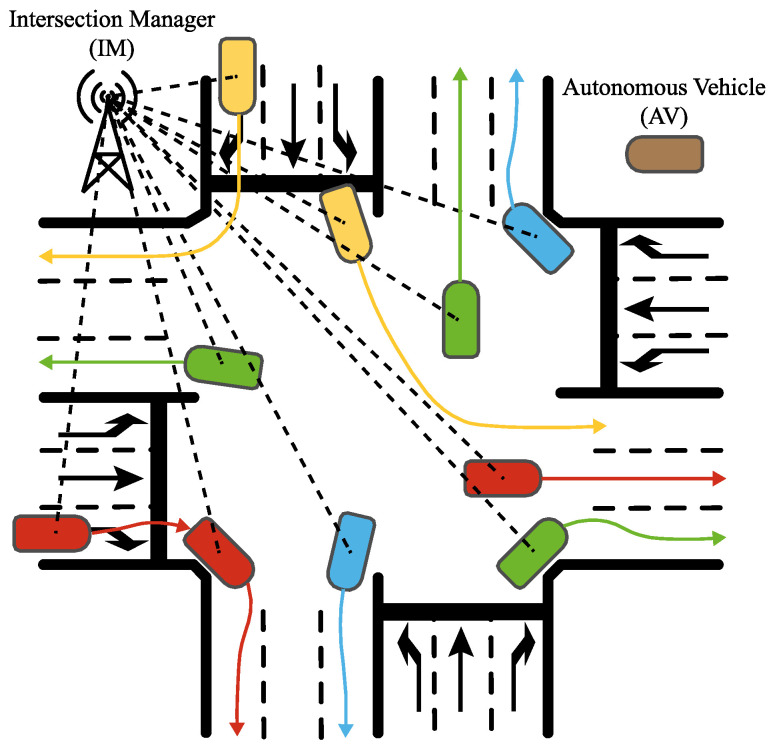
Example of Autonomous Intersection Management (AIM). The Intersection Manager (IM) communicates with the Autonomous Vehicles (AVs) through a wireless communication network and guides them on the action to be taken by each AV. The AIM operates within the IM.

**Figure 2 sensors-22-02217-f002:**
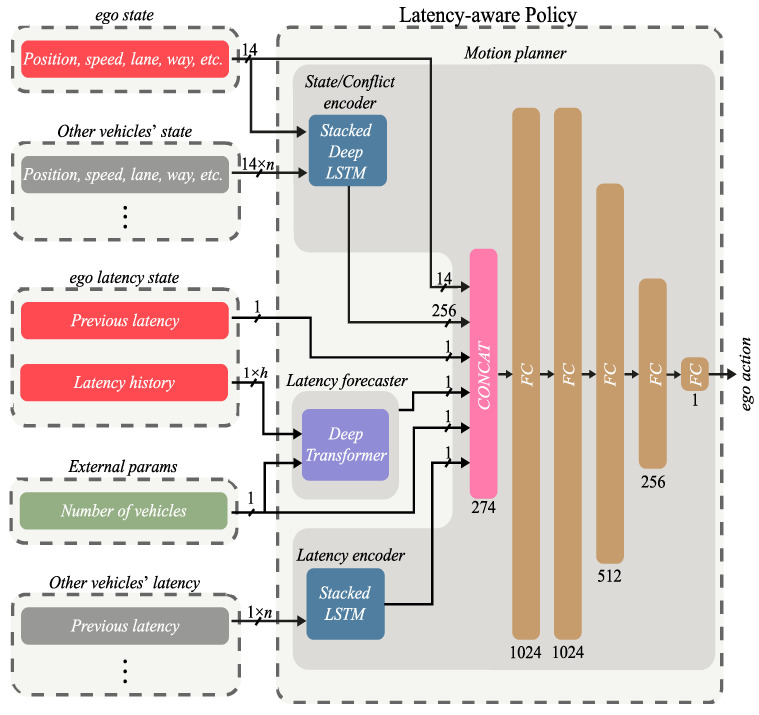
AIM5LA architecture with latency forecaster module based on Deep Transformer model and Latency encoder based on Stacked LSTM network.

**Figure 3 sensors-22-02217-f003:**
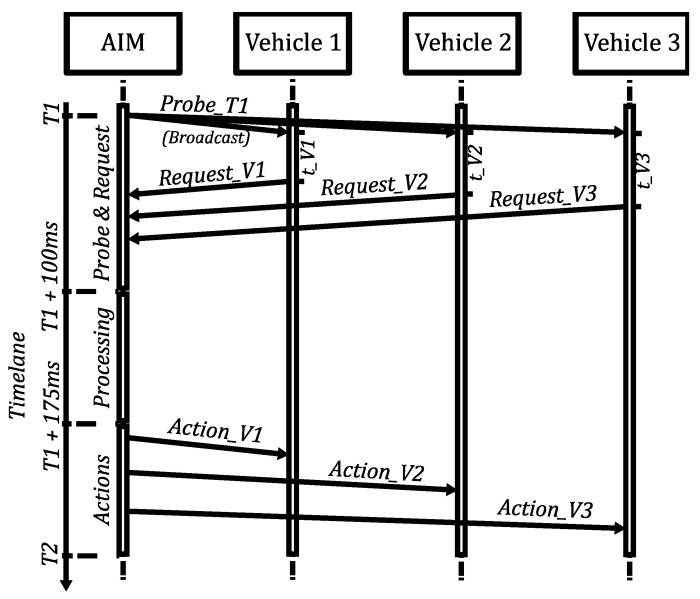
Message diagram proposed for AIM5LA. In this example, there are 3 vehicles to control. The *probe* message is a broadcast message.

**Figure 4 sensors-22-02217-f004:**
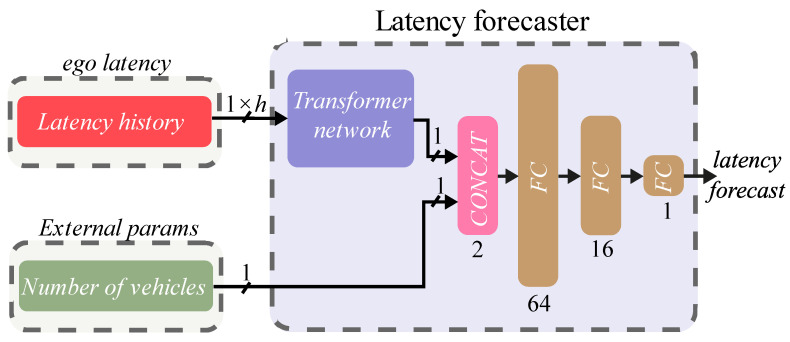
The architecture of the latency forecast module is composed of a transformer network and 3 fully connected layers. The size of the transformer network input depends on the size of the considered window (*h*), which controls how many previous latencies are considered when predicting. The output of the transformer network is concatenated with the variable of the number of simulated vehicles (to be controlled) and fed into the fully connected layer network.

**Figure 5 sensors-22-02217-f005:**
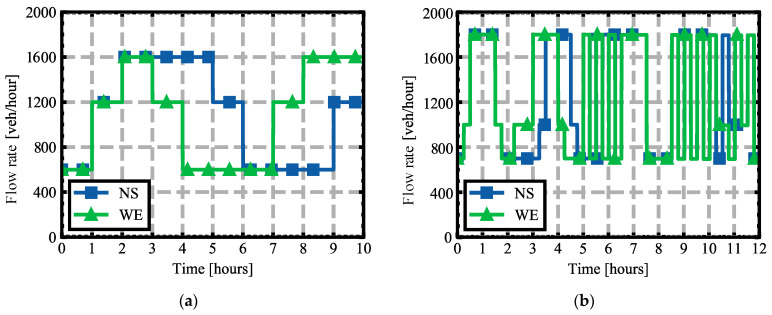
Vehicle flow rate per branch used: (**a**) Training scenario; (**b**) Testing scenario.

**Figure 6 sensors-22-02217-f006:**
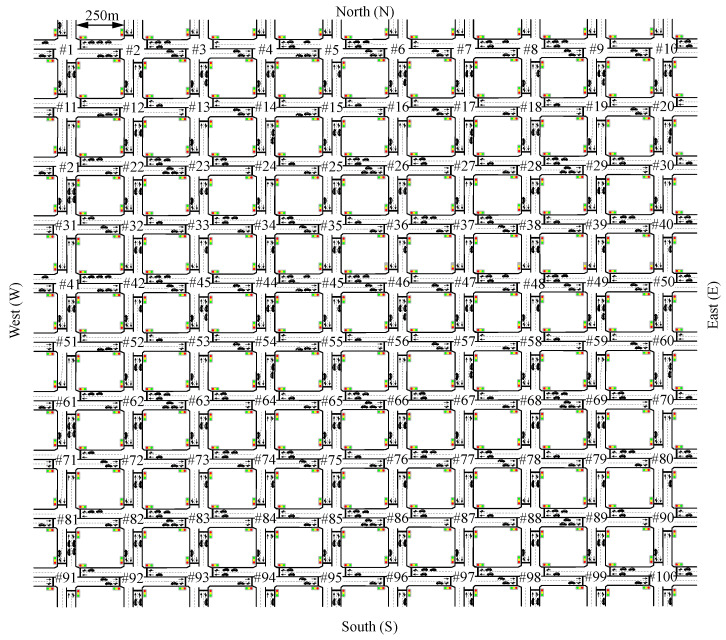
Simulated topology for the testing scenario: Manhattan 10 × 10 network with 250 m between each intersection.

**Figure 7 sensors-22-02217-f007:**
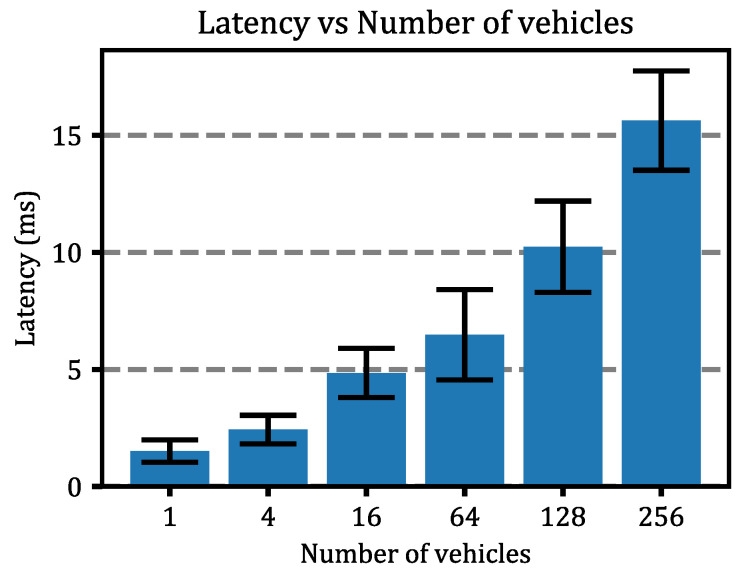
Latency vs. Number of vehicles. It represents the mean and standard deviation values for each group.

**Figure 8 sensors-22-02217-f008:**
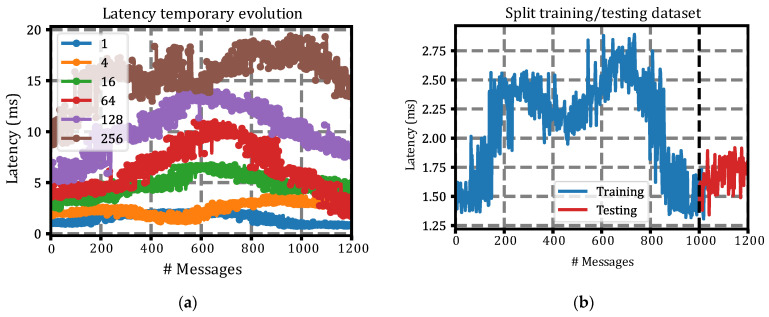
Latency behavior and split: (**a**) Latency temporary evolution for a random vehicle in each simulation group. The legend indicates the number of vehicles simulated simultaneously; (**b**) Example of dataset division. Up to latency sample (message), 1000 was used to form the training dataset, blue line. From the latency sample (message), 1000 was used for the test dataset, red line.

**Figure 9 sensors-22-02217-f009:**
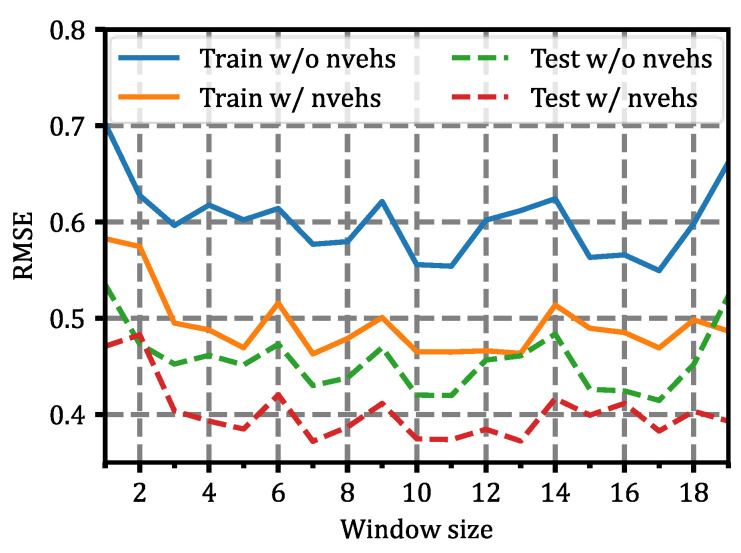
Results of the latency forecast module for both the train and test datasets. The results are shown with the vehicle number variable (w/nvehs) and without it (w/o nvehs) as input parameters in the latency forecaster module.

**Figure 10 sensors-22-02217-f010:**
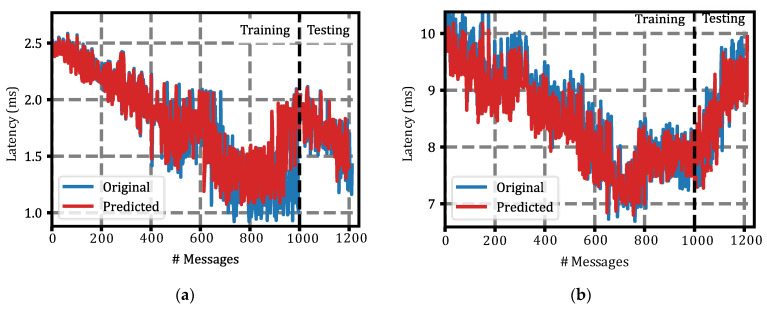
Latency behavior and split: (**a**) Latency temporary evolution for a random vehicle in each simulation group. The legend indicates the number of vehicles simulated simultaneously; (**b**) Example of dataset division. Up to latency sample (message), 1000 was used to form the training dataset, blue line. From the latency sample (message), 1000 was used for the test dataset, red line.

**Figure 11 sensors-22-02217-f011:**
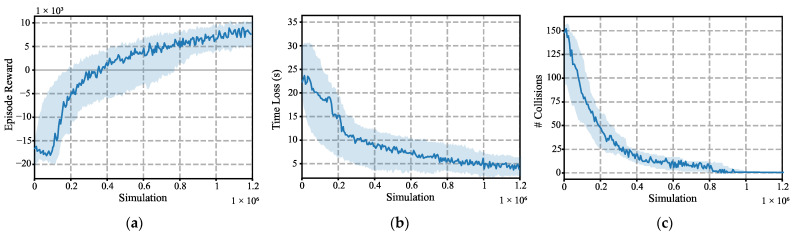
AIM5LA training results. The metrics show the mean (solid line) value and standard deviation (shaded area) of the 3 runs: (**a**) Average episode reward; (**b**) Time Loss (s); (**c**) Number of collisions.

**Table 1 sensors-22-02217-t001:** Summary of Simulator Setup.

Simulator	Parameter	Value
SUMO	Simulation step	0.25 segs
	Flow step	100 veh/h
	Min flow	200 veh/h
	Train duration	5 min/simulation
	Test duration	840 min/simulation
	Scenario	4 branches, 3 lanes/way, and all ways.
	Control distance	100 m
Simu5G	Carrier Frequency	2 GHz
	Bandwidth	20 MHz
	IM Tx Power	40 dBm
	IM antenna gain	12 dB
	IM noise figure	5 dB
	AV antenna gain	3 dB
	AV noise figure	7 dB
	Path loss model	3GPP TP 36.783

**Table 2 sensors-22-02217-t002:** Vehicle distribution and fuel type used.

Vehicle	Distribution (%)	Fuel Type/Electric
Car	30	Gasoline
Car	40	Diesel
Car	20	Electric
Van	5	Diesel
Bus	5	Diesel

**Table 3 sensors-22-02217-t003:** Results of latency simulation.

Number of Vehicles	Mean Latency	Std Latency
1	1.51	0.48
4	2.43	0.61
16	4.85	1.05
64	6.48	1.93
128	10.24	1.95
256	15.63	2.12

**Table 4 sensors-22-02217-t004:** Results of testing scenario.

Algorithm	Time Loss (s)	Collisions	Waiting Time (s)	CO_2_ Emiss. (g)	PMx Emiss. (mg)	Fuel Cons. (mL)	Elect. Cons. (W)
FX30	79.61 ± 8.98	0 ± 0	61.32 ± 7.99	101.78 ± 15.11	78.41 ± 8.54	391.12 ± 64.51	103.48 ± 11.51
FX60	70.16 ± 11.41	0 ± 0	50.71 ± 6.11	89.64 ± 9.87	66.98 ± 7.21	333.74 ± 42.66	99.74 ± 9.63
FX90	72.58 ± 7.42	0 ± 0	55.65 ± 7.05	95.87 ± 8.45	72.54 ± 7.88	351.52 ± 39.98	101.88 ± 9.88
iREDVD [2]	34.97 ± 3.32	0 ± 0	32.23 ± 4.44	53.44 ± 3.22	39.74 ± 6.11	205.25 ± 13.14	66.27 ± 4.48
*adv*.RAIM [4]	5.11 ± 1.24	49.91 ± 9.89	0.25 ± 0.03	25.48 ± 2.69	18.99 ± 2.81	124.47 ± 12.35	33.74 ± 3.81
Andert et al. [25]	4.98 ± 1.18	27.13 ± 3.11	0.24 ± 0.03	26.93 ± 2.33	17.52 ± 1.99	118.94 ± 18.74	31.67 ± 3.39
AIM5LA_v0.1	4.12 ± 1.41	32.01 ± 4.98	0.24 ± 0.02	26.18 ± 1.29	17.82 ± 1.97	119.14 ± 15.29	30.26 ± 3.21
AIM5LA_v0.2	4.89 ± 1.93	3.22 ± 2.61	0.28 ± 0.03	26.96 ± 1.46	18.49 ± 1.89	126.46 ± 18.46	32.34 ± 2.94
AIM5LA	5.42 ± 1.29	0 ± 0	0.31 ± 0.02	27.52 ± 1.97	19.22 ± 2.14	131.87 ± 17.42	34.29 ± 3.66

## Data Availability

Not applicable.

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
