# Peer review of "AIM5LA: A Latency-Aware Deep Reinforcement Learning-Based Autonomous Intersection Management System for 5G Communication Networks"

_sensors, 2022, doi:10.3390/s22062217_

Round 1

Reviewer 1 Report

This paper is very original。 The authors presented in this paper a latency-aware AIM designed to work over a 5G communication network. However, In terms of 5G network delay, I hope the author can further consider the following two problems. 1. How much is the latency of 5G network? Can it be ignored relative to the processing time of AIM5LA? 2. The latency of wireless network fluctuates. Does this fluctuation affect the processing of AIM5LA?

Author Response

Response to Reviewer 1:

Authors’ note: First of all, thanks for your time and comments. Please note that all changes to the manuscript are highlighted with red text. You can find below the responses to your comments, point by point, as well as the proposed solution.

This paper is very original. The authors presented in this paper a latency-aware AIM designed to work over a 5G communication network. However, In terms of 5G network delay, I hope the author can further consider the following two problems.

Authors’ response: Thank you very much for your comments. We appreciate your time and effort to review our paper.

Q: English language and style are fine/minor spell check required.

A: Thank you for your comments. We have corrected several grammatical and syntactical errors.

Q: 1. How much is the latency of 5G network? Can it be ignored relative to the processing time of AIM5LA?

A: Thank you very much for your comment. As it is seen in the paper, the latency of the 5G network depends on the density of devices in an area, i.e., the number of 5G devices that a 5G base station handles. This latency varies from 1.5ms when there is only one vehicle, to 15ms when there are 256 vehicles. Moreover, not only is the latency value important but more important is the standard deviation of the latency, varying from 0.48ms to 2.12ms (Table III, previously Table II). On the other hand, the time relative to AIM5LA processing is already included in the AIM5LA protocol, so latency versus AIM5LA processing time cannot be ignored. As it is seen, if latency is ignored then there are collisions (Table IV, previously Table III) due to latency in the communication system and control offsets. Therefore, the proposed AIM5LA protocol mitigates the latency problem (and its time variation) effectively through deep reinforcement learning.

Q: 2. The latency of wireless network fluctuates. Does this fluctuation affect the processing of AIM5LA?

A: Thank you very much for your comment. This is a very interesting question. The answer is no, latency fluctuations do not affect AIM5LA at all. In fact, AIM5LA is able to learn those latency variations and adapt the control to the predicted future latencies. As it can be seen in Figure 10, the latency prediction module of AIM5LA is able to predict with high accuracy the latency that a vehicle will experience. Moreover, when considering the number of controlled vehicles, the latency predictor module takes this into account to vary the variance of the expected latency, managing to adapt the prediction to the number of simulated vehicles. We have added a sentence highlighting these results and the consequences they have on the article so that potential readers can understand more clearly the results obtained (Last paragraph, section V-A).

Finally, we hope that these responses can help to better understand the contribution of this work to the research community, especially at the intersection of intelligent transportation systems, reinforcement learning, and autonomous driving.

We believe that the improvements introduced in the text thanks to the comments and suggestions of the reviewers clarify the contributions of the proposal, the main benefits of the present work, as well as the relevance of the research presented, resulting in a clearer, more detailed, and more comprehensive document.

Reviewer 2 Report

The RL algorithm which is taken from the other work (under review from the same authors) should be explained in details to find out the relevance and benefits of the latency prediction here, which is the main contribution of this work. Besides, the contribution of this work can be also evaluated much easier in this case.

The relation between the number of vehicles and latency is quite clear and trivial. It would be ineteresting to see how the predicted latency, which is still in scale of "ms" even with high number of vehicles, can higly impact on the number of accidents.

The details of simulation setup are missing. It is suggested to be added.

Conclusion must also include the most important lessons learned.

Author Response

Response to Reviewer 2:

Authors’ note: First of all, thanks for your time and comments. Please note that all changes to the manuscript are highlighted with red text. You can find below the responses to your comments, point by point, as well as the proposed solution.

Authors’ response: Thank you very much for your comments. We appreciate your time and effort to review our paper.

Q: English language and style are fine/minor spell check required.

A: Thank you for your comments. We have corrected several grammatical and syntactical errors.

Q: The RL algorithm which is taken from the other work (under review from the same authors) should be explained in details to find out the relevance and benefits of the latency prediction here, which is the main contribution of this work. Besides, the contribution of this work can be also evaluated much easier in this case.

A: Thank you very much for your comments. We believe that the basic operation of the algorithm on which AIM5LA is based can be well understood by using the following reference (Work presented by the same authors at the NeurIPS 2020 conference, an earlier version can be found at https://www.researchgate.net/publication/357957238_RAIM_Reinforced_Autonomous_Intersection_Management_-_AIM_based_on_MADRL). Given that that reference is available for the audience, we do not include the complete description of algorithm in this paper. Thus, the focus of this work can be kept at how to deal with delays in a scenario with autonomous vehicules and autonomous intersection management in the framework of 5G communications. Nevertheless, to facilitate the readers’ understanding, we have added a paragraph explaining the operation of the base algorithm (adv.RAIM) (paragraph 2, section III). We believe that with the changes introduced in this new version, the document is more focused and concise. Regarding your second suggestion, AIM5LA is compared with an exact implementation of the adv.RAIM algorithm, as well as with different versions of AIM5LA (as an ablation study) (Table IV, previously Table III). From these results, the advantages of considering the latency of the 5G network for vehicular control and different modules (during the ablation study) can be clearly seen, and therefore, the main contributions of this work. We believe that with the comments added during the work (conclusions and last paragraph of chapter 4), the main contributions are more clear, concise, and transparent, as well as the relevance of the research presented for the scientific community.

Q: The relation between the number of vehicles and latency is quite clear and trivial. It would be interesting to see how the predicted latency, which is still in scale of "ms" even with high number of vehicles, can higly impact on the number of accidents.

A: Thank you very much for your comments. Indeed, latency prediction is quite important and in many cases can have significant effects on the number of accidents. In fact, as seen in Table IV (formerly Table III), during the comparison and ablation study (adv.RAIM, Andert et al., AIM5LA v0.1, and AIM5LA v0.2) it can be seen how the other algorithms present collisions during the test scenario. However, AIM5LA is able to adapt the control of the vehicles depending on the number of vehicles (and the latency suffered by each of them). In view of the results, AIM5LA is not affected by the higher communication latency due to scenarios with a large number of vehicles, given the following aspects:

1) AIM5LA can keep learning from the behavior of devices in the 5G network as it has the latency history of the controlled devices. Therefore, it will always be able to adapt to the communication network conditions as well as its requirements and predict the expected latency with high accuracy.

2) AIM5LA integrates the latency knowledge to adapt the control of vehicles, therefore, when there is a higher latency, AIM5LA adjusts the speed followed by vehicles to increase the control margin with vehicles and mitigate accidents.

3) Only up to 256 vehicles have been studied, since due to space limitations, in an urban intersection, not many more vehicles are expected (consider an intersection of 4 branches with 3 lanes, if in each entry lane there were 20 vehicles, the number of vehicles to be controlled would be 240 vehicles). We consider that, under normal conditions, the number of vehicles will be within the limits studied.

Consider that, in the case of latency of 15ms, and vehicles traveling at 50km/h, during the communication delay of 10ms (scenario of 128 vehicles) a vehicle would travel 13.8cm. If this error accumulates for various reasons (accumulation of error sources) at each control interval (every 250ms) it can cause accidents. Moreover, when this communication delay is considered, the control policy that can be found allows for greater robustness and performance.

We find this a very interesting topic and we understand the importance of the results that a study such as the one you highlight can provide. However, we believe that the issue of crashes caused by latency would be better addressed in another complete paper.

Q: The details of simulation setup are missing. It is suggested to be added.

A: Thank you for your feedback. We agree with what you indicate and have added Table I as well as a paragraph in section IV-A that provides more details on the simulation setup.

Q: Conclusion must also include the most important lessons learned.

A: Thank you very much for your comment. We have added a paragraph in the conclusions section that summarizes the most important lessons learned in the development of the work. We believe that, thanks to the modifications made, the document is more focused and clearer, and offers concrete and reproducible results that can benefit the research community in general. In addition, it offers a more focused and accessible view to potential readers.

We believe that the improvements introduced in the text thanks to the comments and suggestions of the reviewers clarify the contributions of the proposal, the main benefits of the present work, as well as the relevance of the research presented, resulting in a clearer, more detailed and more comprehensive document.

Round 2

Reviewer 1 Report

The author has done a lot of work. I have no other questions.

Reviewer 2 Report

Many thanks for the responses. The revised version addresses all the concerns from the reviewer's point of view.